# Caries Experience and Risk Indicators in a Portuguese Population: A Cross-Sectional Study

**DOI:** 10.3390/ijerph20032511

**Published:** 2023-01-31

**Authors:** Eduardo Guerreiro, João Botelho, Vanessa Machado, Luís Proença, José João Mendes, Ana Cristina Manso

**Affiliations:** 1Clinical Research Unit (CRU), Egas Moniz Center for Interdisciplinary Research, Egas Moniz—School of Health and Science, 2829-511 Almada, Portugal; 2Evidenced-Based Hub, Egas Moniz Center for Interdisciplinary Research, Egas Moniz—School of Health and Science, 2829-511 Almada, Portugal

**Keywords:** dental caries, caries experience, epidemiology, prevalence, risk, public health

## Abstract

Oral health surveys are essential for assessing the dental caries experience and to influence national policies. This retrospective cross-sectional study aims to analyze dental caries experience for which dental treatment was sought in a reference university dental hospital at the Lisbon Metropolitan Area between January 2016 and March 2020. Full-mouth examination, and sociodemographic, behavior, and medical information were included. Descriptive analyses and logistic regression analysis were applied to ascertain risk indicators associated with dental caries experience. A final sample of 9349 participants (5592 females/3757 males) were included, aged 18 to 99 years old. In this population, caries experience was 91.1%, higher in female participants. Age (OR = 1.01, 95% CI [1.00–1.02], occupation (OR = 2.94, 95% CI [2.37–3.65], OR = 3.35, 95% CI [2.40–4.67], OR = 2.55, 95% CI [1.66–3.91], for employed, unemployed, and retired, respectively), overweight (OR = 1.52, 95% CI [1.18–1.96]), reporting to have never visited a dentist (OR = 0.38, 95% CI [0.23–0.64], and self-reported week teeth status (OR = 2.14, 95% CI [1.40–3.28]) were identified as risk indicators for the presence of dental caries, according to adjusted multivariable logistic analyses. These results highlight a substantial rate of dental experience in a Portuguese cohort and will pave the way for future tailored oral public health programs in Portugal.

## 1. Introduction

Dental caries is still one of the most common diseases worldwide, affecting 2.3 billion people with the permanent dentition [1], and is characterized by oral biofilm dysbiosis driven by fermentable carbohydrates [1,2,3,4]. Due to pH variations, alternated periods of demineralization and remineralization may exist, and if demineralization predominates, tooth structures will be irreversibly damaged. In the absence of treatment, this lesion progresses to the dentine–pulp interface causing pain and discomfort [2].

Dental caries experience is directly linked to a lower perceived quality of life as well as with considerable economic burden [5,6,7]. If inappropriately managed, people with active dental caries can develop eating problems, tooth loss, and toothache, slower language development in children, as well as absenteeism from school and work [8,9]. Dental caries is unequally distributed among the population, with multiple population groups at higher risk [1,10]. This increased risk includes different factors (i.e., presence of bacteria with cariogenic properties or a cariogenic diet) and indicators (e.g., lifetime exposure to fluoridation in water, oral hygiene habits, dental anxiety, socioeconomic status, education level, smoking habits, among others) [5,10,11].

Robust knowledge of these factors at the populational level contributes to accurate oral health promotion strategies and policies [5,12]. This knowledge partially arises from cross-sectional studies [13], making them of high scientific relevance. Considering the need for aggregated information on caries experience and associated factors [14,15], we retrospectively analyzed a sample of first-incoming patients at a reference Portuguese university dental hospital. Ultimately, we aimed to measure caries experience and identify its risk indicators in the studied population.

## 2. Materials and Methods

### 2.1. Study Design

This retrospective cross-sectional study is a secondary analysis of first-incoming patients at a university dental hospital (Egas Moniz Dental Clinic, Almada, Portugal). This was an uninterrupted data analysis (a non-probability sampling technique) from January 2016 until March of 2020. The end-period time was defined abruptly due to an imposed COVID-19 lockdown by the Portuguese government. This study is reported following the Strengthening the Reporting of Observational Studies in Epidemiology (STROBE) guideline [16,17]. We conducted this research in accordance with the Declaration of Helsinki of 1975, as revised in 2013, and was approved by the Egas Moniz Ethics Committee (ID number 898). Written informed consent was obtained from all participants at the first appointment.

### 2.2. Study Setting and Sample Size

The original data was sourced from an ongoing database of first-incoming patients. In the first appointment, a mandatory triage includes a self-reported health questionnaire, full-mouth clinical observation, and radiographic examinations (along with a panoramic X-ray and/or bitewings). The self-reported questionnaire includes age, sex, education level, employment status, general medical history and medication, smoking habits, and oral hygiene habits. After examination, patient is informed of their status and treatment plan. The participants were observed by dental students, and the final diagnosis validated by qualified clinical assistants.

To be included in this study, patients were required to be willing to participate in the study, to provide written consent, and to be 18 years old or older. Patients were excluded if they were edentulous or had incomplete data. Edentulism was part of the exclusion criteria because it could result from dental caries. Considering this population is reported to have higher prevalence of periodontitis [18,19], this could be a source of overestimation of dental caries experience, particularly the missing teeth component of the Decayed, Missing, and Filled Teeth (DMFT) index. Patients with developmental disorders or special needs were not included in this study because they are followed at the Special Needs Department at the EMDC.

### 2.3. Dependent Variables

Caries experience was measured through the DMF index and was the main dependent variable. The most used dental caries index is the DMF index, which counts the number of DMFT resulting from dental caries. This index captures an individual’s cumulative experience of past and present dental caries, whether untreated (the number of decayed teeth) or treated (filled teeth or missing teeth extracted as a result of dental caries) [2].

### 2.4. Independent Variables

Sociodemographic and behavior information were collected from the self-reported questionnaire. Health determinants and sociodemographic factors included important independent variables for subsequent analysis such as age, sex, education level, and occupation. These variables are common predictors of caries [8,20,21].

Caries experience was used as a dichotomous variable (yes or no). Furthermore, DMF was used as a continuous variable. Sex was divided into two groups: male and female. Age was recorded as a continuous variable (years) and then we used the following age groups to organize the information and realize the descriptive analyses: 18–24; 25–44; 45–64; and ≥65.

Education level was categorized following the 2011 International Standard Classification of Education (ISCED-2011): No education (ISCED 0 level), Elementary (ISCED 1–2 levels), Middle (ISCED 3–4 levels), and Higher (ISCED 5–8 levels) [22].

Occupation of each subject was classified as: student, employed, unemployed, and retired. This classification is the same used by Botelho, Machado [19]; and Machado, Botelho [18].

Smoking habits were defined as non-smoker and active smoker. The group of smoker was further divided into three categories: light smokers (<10 cigarettes per day), medium smokers (10–20 cigarettes per day), and heavy smokers (>20 cigarettes per day). This division was also used by Botelho, Machado [19]’ and Machado, Botelho [18].

Alcohol consumption was registered as a dichotomous variable (yes or no).

Body Mass Index (BMI) was calculated as the ratio of the individual’s body weight to the square of their height. The height of the participants was measured in centimeters, using a hard ruler installed vertically and secured with a stable base. Weight was assessed in kilograms (Kg) using mechanical scales. Four BMI categories were defined using World Health Organization (WHO) criteria [23]: underweight (18.5 kg/m^2^), normal weight (18.5–24.9 kg/m^2^), overweight (25–29.9 kg/m^2^), and obese (≥30 kg/m^2^). Variables about oral heath were adapted following the WHO Oral Health Surveys: Basic Methods [24].

Comorbidity was defined as an occurrence of one or more self-reported systemic disorders including endocrine disorders, blood vascular disorders, orthopedic diseases (arthritis, rheumatoid arthritis), hypertension, and allergy [25]. The number of comorbidities were divided in 4 groups (low—1, moderate—2 or 3, high—4 or 5, and very high—≥6) according to Browne et al. [26].

The time elapsed since last dental consult was classified into five categories (never visited, less than one year, 1–2 years, 3–4 years, 5 years or over). Appointment reasons were classified as routine, aesthetics, pain, functional, or other. Oral hygiene habits were assessed by information on toothbrush frequency (2–3 times/daily, 1 time daily, 2–6 times/weekly, and never), dental floss use, and mouthwash use. The oral self-perception was divided in two groups, Teeth Health and Gums Health, each one classified into five categories (excellent, very good, good, weak, and very weak).

### 2.5. Statistical Analysis

Data analysis was performed using IBM SPSS Statistics version 28.0 for Windows (IBM Corp., Armonk, NY, USA). Descriptive and inferential statistics methodologies were applied. The homogeneity of variance was calculated with Kolmogorov–Smirnov test and Levene’s test.

For variables with more than two independent samples, normal distribution, and homogeneous variance, we use ANOVA I and Tukey HSD as post hoc tests to compare clinical data with sociodemographic variables. The Kruskal–Wallis test and pairwise comparison with Bonferroni correlation are performed when the data are normally distributed and homogeneity of variance is rejected or when the data are not normally distributed. In cases where two independent samples are normally distributed and homogeneity of variance is accepted, we use the parametric Student’s *t*-test. If homogeneity of variance is rejected, we use the parametric Welch test. Mann–Whitney is used when data are not normally distributed.

Logistic regression analysis explored the relationship between dental caries and conceivable risk indicators. Preliminary analyses were performed using univariate models. Next, a multivariate model constructed using variables showing a significance *p* ≤ 0.25 in the univariate model were included in the multivariate stepwise procedure. Among the predictor variables were sex, age (years), education level, occupation, smoking and drinking alcohol habits, BMI, last dental visit, appointment reasons, toothbrush frequency, dental floss use, mouthwash use, tooth and gums health perception, and presence of comorbidities. The contribution of each variable to the model was evaluated by Wald statistics. Interactions were also analyzed for all tested variables. The final reduced model included: occupation (student, employed, unemployed, and retired), BMI (overweight), last dental visit (never), and dental status perception (week). Odds ratio (OR) and 95% confidence intervals (95% CI) were calculated for both univariate and multivariate analyses. The level of statistical significance was set at *p* ≤ 0.05.

## 3. Results

### 3.1. Participant Inclusion and Characteristics

From a total of 9860 incoming patients, 9349 (94.8%) fulfilled the eligibility criteria, while 511 participants were excluded from the study. Among the excluded individuals, 306 (59.9%) were younger than 18 years, 204 (39.9%) were edentulous, and one (0.2%) had an incomplete triage questionnaire.

Regarding the 9349 participants, the majority were female participants (59.8%) and age ranged between 25 and 64 years old (64.3%). Most participants reported having an elementary or middle school education (65.9%) and being employed (53.3%). In addition, 73.8% were not smokers, 52.5% reported alcoholic habits, and 49.6% were overweight and obese. Overall, 52.0% of this sample had at least one comorbidity (Table 1).

Regarding oral health self-reported perception, 51.7% claimed to have seen a dentist in the last year and the most common appointment reason was a functional complaint (46.1%) followed by routine (28.1%) and a pain event (18.9%) (Table 2). About 80.2% reported they brush their teeth 2–3 times a day, yet only 36.7% said they performed interproximal hygiene with dental floss. A few participants (1.8%) considered their teeth to be excellent, while 43.1% and 46.7% considered them good and weak/very weak, respectively. Regarding gum health self-perception, the majority (53.9%) considers their gums to be good.

### 3.2. Dental Caries Experience

Out of the 9349 participants, 8521 (91.1%) had caries experience, of which 59.7% (n = 5090) were female subjects (Table 3). Males had significant higher decayed teeth (*p* < 0.001) and lower filled teeth (*p* < 0.001) than female participants, while no differences were found for missing teeth (*p* = 0.842).

With regard to the age intervals, people ranging 25 and 44 years had the highest average number of decayed teeth (6.9), with a significant difference among the remaining age groups (*p* < 0.001).

The elementary education group (6.9) has more decayed teeth than the higher education group (5.1). The elementary education, middle education, and higher education groups have significantly different decayed tooth rates (*p* < 0.001). The higher education group (5.1%) has fewer decayed teeth. The mean number of decayed teeth in the primary education group is the highest. There is a significant difference between the number of missing teeth between the elementary, middle, and higher education groups (*p* = 0.001), with the higher education group having a lower missing tooth rate (3.5), and groups with no studies (12.5) and elementary studies having a higher missing tooth rate (11.5).

According to this cross-sectional study, of the 8521 participants with past caries experience, 6277 (73.7%) are non-smokers, whereas 2244 (26.3%) smoke. Despite the difference in experience between smokers and non-smokers, there is no statistically significant difference in the mean number of decayed teeth (*p* = 0.644). Although there is no statistically significant difference between smokers and non-smokers regarding the number of dental caries, data suggest that dental caries incidence depends on the type of active smoker. A 49.6% caries rate was observed among active smokers who smoked between 10 and 20 cigarettes a day. Additionally, their mean number of DMFT was higher. Heavy smokers (>20 cigarettes a day) have the highest mean number of DMFT.

Dental caries was estimated to occur in 34.8% of normal-weight people, 29.8% of obese people, and 16.8% of underweight people. Despite normal weight representing the group with the highest caries experience (38.2%), there is a statistically significant difference (*p* = 0.001) between overweight and obese groups regarding decayed and missing teeth.

With a self-reported systemic disorder (51.9%), the results show a high caries rate. Despite the caries experience exceeding 50%, there is no statistically significant difference between the mean number of decayed (*p* = 0.346), missing (*p* = 0.051), or filled teeth (*p* = 0.989).

Oral health care and self-reported perceptions about oral health were associated with higher levels of decayed teeth as well as negative self-perceptions about tooth health (Table 4). A similar pattern has been observed in the mean number of missing teeth, except for those who thought their teeth were excellent, whose mean number of missing teeth was higher. Among these groups, there is a statistically significant difference in decayed teeth, except for the “weak” and “very weak” groups.

### 3.3. Analysis of Risk Indicators

After univariate logistic regression analyses (Appendix A), significant variables were explored with multivariate logistic regression (Table 5). Age was a significant variable for DMFT (OR = 1.01, *p* = 0.018). Occupation also showed significance, with retired people showing the highest risk towards caries (OR = 3.35, *p* < 0.001). Regarding body weight distribution, overweight and obese people showed higher likeliness to present dental caries (OR = 1.52, *p* = 0.001; OR = 1.36, *p* = 0.038, respectively).

People reporting to have never visited a dentist had a significantly lower risk of presenting dental caries (OR = 0.38, *p* < 0.001). Oral health self-perception was also linked to dental caries presence.

## 4. Discussion

This study retrospectively analyzed dental caries experiences in a Portuguese adult population based on both clinical and radiographic examinations. Nine out of ten participants had some level of caries experience at the time of observation, according to the DMFT index. Among the significant risk indicators, age, employment status, body fat based on height and weight, self-perceived teeth status, and frequency of dental check-ups were the most relevant to the prediction of dental caries experience.

Overall, these results are relevant to the studied population based on the characteristics and oral health system in place. The oral healthcare system in Portugal is mainly based upon private practice [2]. In 2005, the Portuguese Public Oral Health Program (PPOHP) launched a “dental voucher” program for children, adolescents, and vulnerable groups [3]. These dental vouchers are then used by patients at primarily private practice clinics, despite existing dental care in the Portuguese National Health System which reveals its insufficiency to respond to population needs. The final application of this research is to serve as a baseline for a different approach to the management of dental caries.

In this study, women were observed with a higher rate of dental caries, yet their caries experience was not statistically different from men, in line with other studies [27,28]; nevertheless, sex differences in caries experience have also been reported [8]. Culture, subsistence systems, dietary patterns, and even hormonal fluctuations can influence caries experiences differently between males and females [29,30,31].

Age was also a significant risk indicator for dental caries experience, expectedly, possibly due to higher exposure to a cariogenic diet [29]. In accordance with literature [30,32,33], age remains a relevant risk indicator and our results are no exception. This link may also be explained by several other factors that could be attributed to ageing such as xerostomia, polypharmacy, functional and cognitive impairment, or an intraoral ecological alteration throughout time [30,32].

Participants’ schooling and employment activity are also revealed to be relevant. Lower education or unemployed participants had higher levels of dental caries and dental caries experience. Our results are consistent with other studies where jobless people had poorer clinically determined oral health compared to the employed [34,35,36]. Occupational environments have a significant impact on oral health [21,36,37,38].

Several factors can harm adults’ oral health, such as stress at work, healthcare policies, and health-insurance companies [34]. Uncertainties about how unemployment affects oral health are yet unanswered, but there are some hypotheses that could explain the reality such as the fact that dental care is considered expensive even for employed adults and that public dental care is almost nonexistent in some countries [34,39]. The Portuguese government has implemented a few policies and programs to improve oral health in the country, including initiatives to increase access to dental care for disadvantaged groups and to promote oral hygiene and preventive care. Despite these efforts in oral health care, more initiatives should be carried out to improve access to oral health [40,41].

Our results also show that body fat based on height and weight measured through the BMI is also linked significantly to the presence of dental caries, particularly with people who are overweight/obese having higher levels, and are consistent with other studies [42,43,44]. It is possible that the increased experience of caries in the overweight and obese groups is due to other factors, such as dietary habits such as the consumption of sugary drinks and foods [42,45]. However, even though the results of this study demonstrate a link between a higher BMI Index and dental caries, a more in-depth understanding of how obesity affects oral health, including dental caries, is necessary because there also findings that suggest an inverse relationship between dental caries experience and obesity [46,47].

These findings demonstrate that people reporting to have never visited a dentist had had a significantly lower risk of presenting dental caries (OR = 0.38, *p* < 0.001). It is important to remind readers that from 9.349 participants of this study, only 112 (1.2%) report that they have never visited a dentist. Appointment motivations may explain this result. Our data show that only 2.620 participants (28.1%) looked for routine appointments. This led us to believe that it was more common to see patients who were “problem-oriented” than those who were “prevention-oriented” and these conclusions are shared with other similar studies [8,48,49] and by the oral health report of the Portuguese Dental Association [50].

The results of this cross-sectional study are useful for providing evidence that dental caries is a disease that is not equally distributed among the population, affecting several population groups.

### Strenghts and Limitations

We have strengths and limitations to consider in our study, which are worth taking into consideration. One of the limitations of this study is related to the study design. This study is observational and therefore hinders any cause-and-effect testing, but it is especially noteworthy that the number of participants was large. There was also a limitation related to the fact that students primarily observed the participants. This limitation was offset by the fact that all diagnostics were validated by qualified teachers.

The lack of control for other potential variables of interest such as exposure to fluoride, salivary flow, or socioeconomic status where most patients declined to provide their socioeconomic status constitutes potential limitations of this study.

Other limitations important to mention are related to the DMFT index. When determining the DMFT index, the mix of decayed, missing, and filled teeth is not considered, nor is it considered whether teeth are lost due to other reasons besides caries. The DMFT index validity is therefore compromised [51]. The DMF does not indicate the need for dental treatment. However, the ratio of decayed teeth to the total number of teeth in the DMF (D/DMF) can be used as an estimate of unmet treatment needs. Similarly, the ratio of filled teeth to the total number of teeth in the DMF (F/DMF) can be interpreted as a measure of a person’s access to dental care. However, we emphasize that radiographic confirmation of dental caries may be seen as an advantage of our clinical confirmation of dental caries, increasing the consistency of our estimate.

The BMI index also has several limitations when it comes to evaluating the risk or experience of dental caries. This index may be a useful tool for assessing overall health and risk of certain diseases, but it should not be used as the sole indicator of dental caries experience. It is important to consider a range of factors, including diet, oral hygiene, and overall health status, when evaluating an individual’s risk of dental caries [45].

Nevertheless, this study is reported upon an international and widely accepted guideline [16,17].

## 5. Conclusions

Our results show a high burden of dental caries experience. Age, occupation, body fat based on height and weight, dental health self-perception, and frequency of dental check-ups were the significant risk indicators. These results will pave the way for future tailored public health programs for dental caries.

## Figures and Tables

**Table 1 ijerph-20-02511-t001:** Sociodemographic, health, and behavior characterization of the participants (n = 9349).

Variable	Sub-Variable	n (%)
Sex	Female	5592 (59.8)
	Male	3757 (40.2)
Age group (years)	18–24	1867 (20.0)
	25–44	2907 (31.1)
	45–64	3101 (33.2)
	≥65	1474 (15.8)
Education	Without studies	38 (0.4)
	Elementary	2668 (28.5)
	Middle	3492 (37.4)
	Higher	3151 (33.7)
Occupation	Student	1616 (17.3)
	Employed	4980 (53.3)
	Unemployed	1083 (11.6)
	Retired	1670 (17.9)
Smoking habits	Smoker	2453 (26.2)
	Non-smoker	6896 (73.8)
Active smokers (Cigarettes per day)	Light	1132 (46.1)
	Medium	1306 (53.2)
	Heavy	15 (0.6)
Alcohol consumption	No	4438 (47.5)
	Yes	4911 (52.5)
BMI (Kg/m^2^)	Underweight	1035 (11.1)
	Normal weight	3683 (39.4)
	Overweight	2960 (31.7)
	Obese	1671 (17.9)
Comorbidity	No	4488 (48.0)
	Yes	4861 (52.0)
Number of comorbidities	Low	2559 (27.4)
	Moderate	1866 (20.0)
	High	349 (3.7)
	Very High	87 (0.9)

Abbreviations: BMI—Body Mass Index; n—number of participants.

**Table 2 ijerph-20-02511-t002:** Oral health care, dental caries experience, and self-reported perception about oral health condition descriptive data (n = 9349).

Variables		n (%)
Last dental visit	Never	112 (1.2)
	<1 year	4.835 (51.7)
	1–2 years	1.401 (15.0)
	3–4 years	1.450 (15.5)
	≥5 years	1.551 (16.6)
Appointment reasons	Routine	2.628 (28.1)
	Aesthetics	408 (4.4)
	Pain	1.768 (18.9)
	Functional	4.312 (46.1)
	Other	233 (2.5)
Toothbrush frequency	2–3 times/daily	7.496 (80.2)
	1 time/daily	1.550 (16.6)
	2–6 times/weekly	156 (1.7)
	Never	147 (1.6)
Dental floss usage	No	5.917 (63.3)
	Yes	3.432 (36.7)
Dental caries experience	No (DMFT = 0)	204 (2.2)
	Yes (DMFT > 0)	9.145 (97.8)
	DT	8.521 (91.1)
	MT	6.730 (72.0)
	FT	6.365 (68.1)
Gum bleeding	No	5.143 (55.0)
	Yes	4.206 (45.0)
Teeth health perception	Excellent	166 (1.8)
	Very good	790 (8.5)
	Good	4.030 (43.1)
	Weak	2.886 (30.9)
	Very weak	1.477 (15.8)
Gums health perception	Excellent	333 (3.6)
	Very good	1.033 (11.0)
	Good	5.041 (53.9)
	Weak	2.291 (24.5)
	Very weak	651 (7.0)

Abbreviations: n—number of participants; D: Decayed Teeth; M: Missing teeth; F: Filled Teeth.

**Table 3 ijerph-20-02511-t003:** Dental caries data (presented as mean and standard deviation) as function of sociodemographic, health, and behavior factors (n = 9349).

Variable		n (%)	DT	MT	FT	DMFT
Sex	Female	5.090 (59.7)	5.8 (4.3) ^a^	6.6 (7.4) ^a^	3.3 (3.6) ^a^	15.7 (8.2) ^a^
	Male	3.431 (40.3)	6.3 (4.8) ^b^	6.5 (7.4) ^a^	2.7 (3.3) ^b^	15.5 (8.2) ^a^
Age group (years)	18–24	1.496 (17.6)	4.6 (4.5) ^a^	0.7 (1.5) ^a^	1.9 (2.6) ^a^	7.3 (6.3) ^a^
	25–44	2.702 (31.7)	6.9 (4.9) ^b^	3.5 (4.5) ^b^	3.6 (3.7) ^b^	14.1 (7.0) ^b^
	45–64	2.932 (34.4)	6.2 (4.1) ^c^	9.4 (7.2) ^c^	3.7 (3.8) ^c^	19.2 (6.7) ^c^
	≥65	1.391 (16.3)	5.5 (4.1) ^d^	13.9 (7.9) ^d^	2.1 (2.8) ^a^	21.5 (7.0) ^d^
Education	Elementary	2.544 (29.9)	6.9 (4.9) ^a^	11.5 (8.3) ^a^	1.9 (2.7) ^a^	20.3 (7.5) ^a^
	Middle	3.176 (37.3)	6.0 (4.5) ^b^	5.5 (6.5) ^b^	3.1 (3.5) ^b^	14.7 (7.9) ^b^
	Higher	2.764 (32.4)	5.1 (4.1) ^c^	3.5 (4.9) ^c^	4.0 (3.9) ^c^	12.6 (7.3) ^c^
	Without studies	37 (0.4)	6.5 (4.8) ^abc^	12.5 (7.9) ^a^	1.4 (2.5) ^a^	20.5 (7.1) ^a^
Occupation	Student	1.255 (14.7)	4.1 (4.2) ^a^	0.9 (2.2) ^a^	2.2 (2.8) ^a^	7.2 (5.5) ^a^
	Employed	4.661 (54.7)	6.4 (4.5) ^b^	5.7 (6.2) ^b^	3.7 (3.8) ^b^	15.8 (7.3) ^b^
	Unemployed	1.024 (12.0)	7.4 (5.1) ^c^	8.3 (7.9) ^c^	2.7 (3.4) ^b^	18.4 (7.5) ^c^
	Retired	1.581 (18.6)	5.6 (4.1) ^d^	13.4 (8.0) ^d^	2.2 (2.8) ^a^	21.2 (7.1) ^d^
Smoking habits	Non-smoker	6.277 (73.7)	6.0 (4.6) ^a^	7.1 (7.6) ^a^	2.9 (3.4) ^a^	15.9 (8.4) ^a^
	Smokers	2.244 (26.3)	6.0 (4.4) ^a^	5.2 (6.4) ^b^	3.6 (3.7) ^b^	14.7 (7.7) ^b^
Active smokers	Light	1.014 (41.3)	5.6 (4.5) ^a^	4.1 (6.0) ^a^	3.4 (3.7) ^a^	13.2 (7.6) ^a^
	Medium	1.216 (49.6)	6.2 (4.3) ^b^	6.0 (6.6) ^b^	3.7 (3.6) ^b^	16.0 (7.5) ^b^
	Heavy	14 (0.6)	6.5 (4.9) ^b^	7.5 (6.5) ^ab^	5.0 (5.2) ^ab^	19.1 (5.2) ^bc^
BMI (Kg/m^2^)	Underweight	913 (10.7)	5.6 (4.7) ^a^	5.3 (7.0) ^a^	3.1 (3.5) ^a^	14.1 (8.4) ^a^
	Normal weight	3.252 (38.2)	5.7 (4.6) ^a^	4.9 (6.7) ^a^	3.1 (3.5) ^a^	13.7 (8.2) ^a^
	Overweight	2.786 (32.7)	6.3 (4.4) ^b^	7.6 (7.6) ^b^	3.2 (3.6) ^a^	17.1 (7.7) ^b^
	Obese	1.570 (18.4)	6.3 (4.5) ^b^	9.0 (7.8) ^c^	2.7 (3.3) ^b^	18.1 (7.9) ^c^
Comorbidity	No	4.102 (48.1)	6.0 (4.5) ^a^	6.7 (7.5) ^a^	3.1 (3.5) ^a^	15.8 (8.1) ^a^
	Yes	4.419 (51.9)	5.9 (4.6) ^a^	6.4 (7.3) ^a^	3.1 (3.5) ^a^	15.4 (8.3) ^a^

Data are mean (standard deviation). Different letters indicate statistically different mean values (Tukey HSD test, *p* < 0.05). Abbreviations: BMI—Body Mass Index; DMFT: Decayed, Missing, Filled Teeth index; DT: decayed teeth; MT: missing teeth; FT: filled teeth.; DMFT—Decayed, Missing, and Filled Teeth index; n—number of participants. Statistical analysis for a significance level *p* < 0.05.

**Table 4 ijerph-20-02511-t004:** Oral health care and self-reported perception about oral health condition (n = 9349).

Variable	Sub-Variable	n (%)	DT	MT	FT	DMFT
Last dental visit	<1 year	4.374 (51.3)	5.9 (4.5) ^a^	6.0 (7.1) ^a^	3.1 (3.5) ^a^	15.0 (8.2) ^a^
	1–2 years	1.291 (15.2)	6.1 (4.5) ^a^	6.4 (7.3) ^ab^	3.1 (3.5) ^a^	15.6 (8.2) ^b^
	3–4 years	1.320 (15.5)	6.1 (4.5) ^a^	6.7 (7.5) ^b^	3.1 (3.5) ^a^	15.8 (8.0) ^cb^
	≥5 years	1.444 (16.9)	6.2 (4.6) ^a^	8.3 (7.8) ^cd^	3.0 (3.5) ^a^	17.4 (8.0) ^d^
	Never	92 (1.1)	5.5 (4.9) ^b^	7.5 (8.7) ^abd^	2.6 (3.2) ^a^	15.5 (9.1) ^abcd^
Toothbrush frequency	2–3 times/daily	6.802 (72.8)	6.0 (4.6) ^a^	6.1 (7.1) ^a^	3.1 (3.5) ^a^	15.1 (8.2) ^a^
	1 time/daily	1.436 (15.4)	6.0 (4.4) ^a^	8.2 (8.1) ^b^	3.0 (3.4) ^ab^	17.1 (8.2) ^b^
	2–6 times/weekly	140 (1.5)	5.9 (4.4) ^a^	9.6 (8.4) ^c^	2.7 (3.5 ^bc^	18.2 (8.1) ^bc^
	Never	143 (1.5)	6.9 (4.9) ^a^	10.8 (8.4) ^c^	2.4 (3.3) ^c^	20.0 (8.0) ^c^
Teeth health perception	Excellent	131 (1.5)	4.6 (4.5) ^a^	3.1 (5.7) ^a^	2.5 (3.1) ^a^	10.2 (7.9) ^a^
	Very good	671 (7.9)	5.3 (4.7) ^b^	2.8 (5.0) ^a^	2.6 (3.3) ^a^	10.6 (7.6) ^a^
	Good	3.625 (42.5)	5.9 (4.6) ^c^	5.7 (7.1) ^b^	3.0 (3.4) ^b^	14.5 (8.2) ^b^
	Weak	2.711 (31.8)	6.2 (4.4) ^d^	7.9 (7.5) ^c^	3.3 (3.5) ^c^	17.3 (7.7) ^c^
	Very weak	1.383 (16.2)	6.3 (4.5) ^d^	8.9 (7.8) ^d^	3.2 (3.8) ^bc^	18.5 (7.6) ^d^
Gums health perception	Excellent	287 (3.4)	5.5 (4.6) ^a^	4.2 (5.8) ^a^	2.8 (3.4) ^a^	12.5 (8.0) ^a^
	Very good	905 (10.6)	5.6 (4.8) ^a^	4.1 (4.3) ^a^	2.7 (3.3) ^ab^	12.4 (8.3) ^a^
	Good	4.600 (54.0)	6.0 (4.5) ^b^	6.7 (7.4) ^b^	3.1 (3.5) ^ac^	15.7 (8.2) ^b^
	Weak	2.122 (24.9)	6.0 (4.4) ^b^	7.3 (7.6) ^c^	3.2 (3.5) ^c^	16.5 (7.9) ^c^
	Very weak	607 (7.1)	6.2 (4.5) ^b^	8.5 (7.6) ^d^	3.4 (3.8) ^c^	18.1 (7.6) ^d^

Data are mean (standard deviation). Different letters indicate statistically different mean values (Tukey HSD test, *p* < 0.05). Abbreviations: BMI—Body Mass Index; DMFT: Decayed, Missing, Filled Teeth index; DT: decayed teeth; MT: missing teeth; FT: filled teeth.; DMFT—Decayed, Missing and Filled Teeth index; n—number of participants. Statistical analysis for a significance level *p* < 0.05.

**Table 5 ijerph-20-02511-t005:** Multivariate logistic regression analysis (final reduced model *) towards the outcome variable ‘caries presence’ (n = 9349).

Variable		OR (95% CI)	*p*
Age		1.01 (1.00–1.02)	0.018
		-	<0.001
Occupation	Student	1	-
	Employed	2.94 (2.37–3.65)	<0.001
	Unemployed	3.35 (2.40–4.67)	<0.001
	Retired	2.55 (1.66–3.91)	<0.001
BMI (kg/m^2^)	Underweight	1	-
	Normal weight	1.04 (0.83–1.29)	0.756
	Overweight	1.52 (1.18–1.96)	0.001
	Obese	1.36 (1.02–1.81)	0.038
Last dental visit	<1 year	1	-
	1–2 years	1.13 (0.91–1.42)	0.266
	3–4 years	0.90 (0.73–1.11)	0.337
	≥5 years	0.99 (0.79–1.25)	0.932
	Never	0.38 (0.23–0.64)	<0.001
Teeth health perception	Excellent	1	-
	Very good	1.47 (0.95–2.27)	0.084
	Good	1.65 (1.10–2.48)	0.015
	Weak	2.14 (1.40–3.28)	<0.001
	Very weak	1.79 (1.13–2.82)	0.013

* The model was statistically significant, χ^2^(15) = 417.443, *p* < 0.001, explained 9.7% (Nagelkerke R^2^) of the variance, and correctly classified 91.1% of cases. Abbreviations: n—number of participants; BMI—Body Mass Index; CI—Confidence Interval; OR—Odds Ratio.

## Data Availability

All data generated or analyzed during this study are included in this article. Further enquiries can be directed to the corresponding author.

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
