# Peer review of "Caries Experience and Risk Indicators in a Portuguese Population: A Cross-Sectional Study"

_ijerph, 2023, doi:10.3390/ijerph20032511_

Round 1

Reviewer 1 Report

Dear Authors, your cross-sectional study is quite interesting because it involves almost 4 years of patients, from 2016 until 2020. It has a great amount of data with over 9000 patients reviewed, what gives a lot of information and, as you point out on your article, is a cross sectional study, it has a statistical power due to the high number of patients.

The structure of the article is correct, as well as the statistical analysis and the material and methods is well described. But I feel I miss the number of investigators that made the dental revisions and who were they? You have to point out who were the investigators that made the dental revisions? I imagine that there were the dental students in the university, is this so, you have to point it out, and there you have a limitation on your study.

The discussion is well done, as well as the conclusion, and you should point out in the strengths and limitations that you compare the study with other studies but not from Portugal.

With this corrections this could be a great article.

Author Response

Dear Reviewer,
Thank you for your comments, suggestions and for the encouraging words.
In the following paragraphs you can find a point-by-point response to each one of your comments.

1.The structure of the article is correct, as well as the statistical analysis and the material and methods is well described. But I feel I miss the number of investigators that made the dental revisions and who were they? You have to point out who were the investigators that made the dental revisions? I imagine that there were the dental students in the university, is this so, you have to point it out, and there you have a limitation on your study.

Our answer:

We appreciate your remark. Participants were primarily observed by dental students, but all diagnostics were corroborated by qualified teachers. These details were added to the Methods section, and discussed in the limitations of the study.

2. The discussion is well done, as well as the conclusion, and you should point out in the strengths and limitations that you compare the study with other studies but not from Portugal.

Our answer:

We thank you for pointing this out. We have indeed discussed this in the following paragraph (lines 292-299) as follows:
“These findings demonstrate that people reporting to have never visited a dentist had had a significantly lower risk of presenting dental caries (OR = 0.38, p<0.001). Is important to remind that from 9.349 participants of this study, only 112 (1.2%) report that never visited a dentist. Appointment motivations may explain this result. Our data show that only 2.620 participants (28.1%) looked for routine appointments. This led us to believe that it was more common to see patients who were "problem-oriented" than those who were "prevention-oriented" and these conclusions are shared with other similar studies [8, 48, 49] and by the oral health report of the Portuguese Dental Association [50].”

3.With this corrections this could be a great article.

Our answer:

We would like to thank this reviewer for the encouraging words.

Thank you 

Reviewer 2 Report

Dear Authors!

First, please let me congratulate to the nice work you have presented, but also please let me raise a few questions and issues?

Was the original „chief complaint” recorded for the patients, included for the study? As it is mentioned, all the patients involved were „first-incoming patients”, it would be essential to know the reason of visit to the Hospital. Missing this information might cause a bias in the whole study sample.

Also I would like to ask, if the presence of periodontitis has been recorded or not? Has there been any diagnosis regarding the patients periodontal status? This might have influenced the number of missing teeth, and also indirectly the number of decayed and filled teeth (as the tooth could be lost already due to periodontal disease).

As known, BMI and age are directly linked with periodontal disease, all of these factors can be biased missing this information.

Please specify and list all exclusion criteria, I only see total edentulism and missing of data in the questionnaire? Any general medical disease, developmental disorder (for example amelogenesis imperfecta, etc) have been used for exclusion, or those patients have been included?

Please describe, why DMFT index has been used, instead of DMFS index? Also please specify, if the DMFT index has been taken on all teeth of a patient, or only on the Ramfjörd teeth?

Please describe, how many patients have been excluded because of not signing the written informed consent. As you state, that a written informed consent was obtained from all participants, I assume, there might be patients, not signing the statement, and not willing to participate, but I do not see this number in the excluded description.

Author Response

Dear Reviewer,
Thank you for your comments, suggestions and for the encouraging words.
In the following paragraphs you can find a point-by-point response to each one of your comments.

1. Was the original „chief complaint” recorded for the patients, included for the study? As it is mentioned, all the patients involved were „first-incoming patients”, it would be essential to know the reason of visit to the Hospital. Missing this information might cause a bias in the whole study sample.

Our answer:

We appreciate this question, however we did include the “chief complaint” named as “Appointment reasons”. We have also discussed this initially.

2. Also I would like to ask, if the presence of periodontitis has been recorded or not? Has there been any diagnosis regarding the patients periodontal status? This might have influenced the number of missing teeth, and also indirectly the number of decayed and filled teeth (as the tooth could be lost already due to periodontal disease). As known, BMI and age are directly linked with periodontal disease, all of these factors can be biased missing this information.

Our answer:

We appreciate pointing this out. Unfortunately, periodontal screening was based on the PSR which is used only for screening and not for diagnosis. For this reason we could not add such information. Nevertheless, we are now working on overcoming this limitation in a future study of this kind.

3. Please specify and list all exclusion criteria, I only see total edentulism and missing of data in the questionnaire? Any general medical disease, developmental disorder (for example amelogenesis imperfecta, etc) have been used for exclusion, or those patients have been included?

Our answer: 

Patients with developmental disorders or special needs are not observed at the triage appointment, but rather in a department of Special Dentistry Care. For this reason those patients do not take part in this study.

4. Please describe, why DMFT index has been used, instead of DMFS index? Also please specify, if the DMFT index has been taken on all teeth of a patient, or only on the Ramfjörd teeth?

Our answer:

All teeth were examined and we opted for DMFT because our research focus was the patient-level rather than the tooth-level. In detail, all variables relate to the patient-level and it could be biased to assess a DMF based on the tooth surface.

5. Please describe, how many patients have been excluded because of not signing the written informed consent. As you state, that a written informed consent was obtained from all participants, I assume, there might be patients, not signing the statement, and not willing to participate, but I do not see this number in the excluded description.

Our answer:

Thank you for this valid question. In fact, the used data is sourced from a pre-existing and pre-approved registered clinical database where patients do consent to have their information registered. This concept is not new, as we can find similarly in other international databases such as the UK biobank. It is a prerequisite, and for this reason we had no such reason for exclusion.

Thank you